# Semantic Component Association within Object Classes Based on Convex Polyhedrons

**Petra Đurović *,†** , **Ivan Vidović** and **Robert Cupec †**

Faculty of Electrical Engineering, Computer Science and Information Technology Osijek, 31000 Osijek, Croatia; ivan.vidovic@ferit.hr (I.V.); robert.cupec@ferit.hr (R.C.)

* Correspondence: petra.durovic@ferit.hr; Tel.: +385-98-175-1176

† These authors contributed equally to this work.

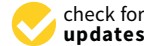

**Featured Application: The application of the proposed work is in robotics. If a certain robot operation is defined for a part of a particular object, it can be transferred to other class instances by applying the proposed method for semantic association of the object components.**

**Abstract:** Most objects are composed of semantically distinctive parts that are more or less geometrically distinctive as well. Points on the object relevant for a certain robot operation are usually determined by various physical properties of the object, such as its dimensions or weight distribution, and by the purpose of object parts. A robot operation defined for a particular part of a representative object can be transferred and adapted to other instances of the same object class by detecting the corresponding components. In this paper, a method for semantic association of the object's components within the object class is proposed. It is suitable for real-time robotic tasks and requires only a few previously annotated representative models. The proposed approach is based on the component association graph and a novel descriptor that describes the geometrical arrangement of the components. The method is experimentally evaluated on a challenging benchmark dataset.

**Keywords:** component association; semantic segmentation; part recognition

## 1. Introduction

One of the trends in robotics is to reduce the need for robot programing by allowing a robot to learn certain tasks from a human instructor. One approach to this problem is kinesthetic training of a robot, where a human manually guides a robot manipulator to perform certain action and then the robot applies the learned action to solve a practical task [1,2]. An advanced version of such training would be to define a robot action for a particular instance of an object class, referred to in this paper as a representative object, and apply an algorithm that would adapt this action to the other instances of the same class. In order to achieve this capability, the considered algorithm must associate the components of the representative object relevant for a particular task with the corresponding segments of the other instances of the same object class. The problem addressed in this paper is how to associate the components of different objects that have the same purpose. Since the target application field considered in this paper is robotics, we define components as regions of the object's surface that could potentially represent contact surfaces between the object and a robot tool when performing some task. For example, if the task is to carry a mug, it should be grasped by the handle. For a light bulb changing task, a robot should be able to recognize the light bulb. A usual approach for semantic segmentation of objects is to train the algorithm on manually annotated training and validation datasets and test it using a test dataset. Since the effort of manually annotating training data is time and energy consuming, the motivation for this paper was minimization of such labor. Without any prior knowledge about

the purpose of a particular object class, the only cues that can be used to solve this problem are the similarity of the shapes of components and their spatial arrangement. The real-time execution of the algorithm is crucial for practical robotic application.

*1.1. The Problem and the Contributions*

The problem addressed in this paper is more precisely stated as follows. Let us consider a database of 3D models of objects belonging to the same class, represented by triangular meshes. There is a number of such datasets that are publicly available. An algorithm selects a small subset of representative objects from this dataset. On each representative object, a human expert annotates the target component specific for a certain task. According to these annotations, the algorithm identifies the semantically corresponding component of each of the remaining models in the database, referred to in this paper as query objects. The proposed approach enables easy and fast expansion of the existing model database with new query objects. In an ideal case, the annotation of a component on a single representative object should be sufficient for identification of all corresponding components in a given object class. However, certain object classes can comprehend objects whose components differ significantly in their shape, size, and position. Therefore, often more than one annotated representative object is required. The focus of our research is computational efficiency, required for practical applications in robotics. Our goal is to develop a method that identifies the target component on a newly perceived object and add this object to the existing database in a few seconds.

The approach presented in this paper is based on the detection of convex and concave surfaces, referred to in this paper as segments. Components are either represented by one or multiple segments. A selected component of the representative object is associated with the corresponding segments of the other instances of the same object class by constructing a component association graph (CAG). Graph nodes represent all segments of all models from the model database. The nodes are interconnected by edges. The weight of each edge represents a measure of the likelihood that these two segments belong to object components that have the same purpose. This measure is comprised of segment size, shape, position, and neighborhood similarity. The proposed similarity measures are based on the convex template instance (CTI) descriptor, proposed in [3], which describes object segments by approximating their shape with convex polyhedrons. A neighborhood similarity is defined by a novel descriptor, named the topological relation descriptor (TRED), which describes the topological relations between two segments in the model, which is also based on the CTI descriptor. Three methods for associating the selected component of the representative models with the segments of the other instances of the same object class using the CAG are proposed in this paper. The direct segment association method associates each segment according to its nearest neighbor in the CAG. The object-constrained association method associates the segments of the query object with each representative objects using a greedy search and computes the matching score between the query object and all representative object. The associations established between the query object and the representative object with the highest matching score are taken as the final result. The MST-based association method associates the query object segments with the representative object segments based on a minimum spanning tree (MST). As the result of any of these three methods, the selected component of the representative model defined for a certain task is associated with the corresponding segments of all models from the model database.

Accordingly, the following contributions of this paper are proposed.

1. A novel computationally efficient approach for establishing associations between components of an object of a given class, based on the component association graph.
2. A novel topologicalrelation descriptor (TRED), which describes the geometrical arrangement of components in a 3D object model.

*1.2. Paper Overview*

The paper is structured as follows. In Section 2, the related research is presented. Sections 3–6 describe the proposed methodology. Section 3 provides a formal problem definition, an overview of

the proposed approach and an explanation of the CAG. In Section 4, an approach for measuring the semantic correspondence likelihood, assigned to the edges of the component association graph, is provided. In Section 5, three methods for the final association between the segments and a target component, based on the CAG, are proposed. Section 6 describes a method for selecting the representative objects. An experimental evaluation and a discussion of the results are given in Section 7. The paper is concluded in Section 8.

## 2. Related Research

Semantic segmentation in 3D was recently established as an important task in various applications: autonomous driving, human-machine interaction, object manipulation, manufacturing, geometric modeling, and reconstruction of 3D scenes, just to name a few. To facilitate the development of 3D shape understanding, the ShapeNet challenge [4] for semantic 3D shape segmentation on a large-scale 3D shape database was proposed. In this challenge, ShapeNet Parts, a subset of 16 classes from the ShapeNet database [5], was used, which is also used in the experimental analysis reported in this paper. The PointCNN method introduced in the ShapeNet challenge was later improved in [6]. It represents a generalization of the typical convolutional neural network (CNN) architecture for feature learning from point clouds. The segmentation accuracy of the PointCNN, experimentally evaluated on ShapeNet Parts dataset, outperformed 14 other methods in segmenting objects belonging to seven classes, while it achieved comparable accuracy in segmentation of objects from the other nine classes. The CNN architectures have been shown to be the most accurate and efficient approaches in a recent review of deep learning techniques applied to semantic segmentation, given in [7]. The best accuracy of object segmentation achieved by seven deep learning methods reported in [4] was between 63 and 96%, depending on the object class.

However, manual annotation of large datasets required for training neural networks is a time-consuming and delicate problem. Therefore, Yi et al. [8] proposed a novel active learning method capable of segmenting massive geometric datasets into accurate semantic regions that grants a good accuracy vs. efficiency trade-off. The ShapeNet Parts dataset was annotated using this approach. The goal of this approach (reducing human work required for the annotation of large datasets) is also the main motivation of the component association method proposed in this paper. The framework [8] achieved the annotation of large-scale datasets by cycling between manually annotating the regions, automatically propagating these annotations across the rest of the shapes, manually verifying both human and automatic annotations, and learning from the verification results to improve the automatic propagation algorithm. Although the approach proposed in [8] included the propagation of component labels from a small object set to a larger set, the focus of their research was on the whole iterative annotation process. A kind of continuation of the research [8] was given in [9]. The method also propagates labels from a small subset to a large dataset by global optimization analogous to the approach proposed in [10], which is based on conditional random fields. Our research, on the other hand, focuses on efficient detection of the target component on a query object given a small annotated set of representative objects, which could allow database expansion in real time.

An automatic approach to achieve semantic annotation of 3D models was proposed in [11]. The approach extracts concave and convex features as the cues for object decomposition into structural parts. By analyzing the position, shape, size, and configuration of the structural parts, the semantic category is assigned to each of them. The proposed methodology resembles the approach proposed by our paper, but it is applied in the semantic annotation of architectural buildings; therefore, the descriptors are adapted to that application, e.g., relative height, volume, dimension ratio, form mode, etc. The final assignment of the semantic label to a part in [11] was performed by the decision tree and the adapted support vector machine, while in this paper, three variants of assigning the label based on the CAG are proposed.

Analogous to the related methods discussed in this section, our method also performs segmentation of complete 3D object models. However, in order to use the algorithm proposed

in this paper in a practical robotic application, a full 3D model of a given query object must be inferred from sensor data. The Pixel2Mesh deep neural network [12], AtlasNet [13], and occupancy network (ONet) [14] reconstruct full 3D object models from RGB images. AtlasNet and ONet can also receive 3D point clouds as the input. The generative shape proposal network (GSPN) [15] processes point clouds for the purpose of solving the semantic segmentation problem. This approach deals with cluttered real indoor scenes, partial point clouds, and point clouds of part-segmented 3D models, which represent real case scenarios in robotic applications.

## 3. Overview of the Proposed Approach

Let $M$ be a set of 3D models of objects belonging to the same semantic class. This model set is referred to in this paper as a model database. It contains 3D models of objects represented by triangular meshes $P_k, k = 1, ..., n_M$. The considered algorithm should select a small set $R$ of representative objects and present them to a human expert, which is asked to annotate a component relevant for a particular task on every object in this set. This annotation assumes the selection of a subset of mesh vertices of each mesh $P_r \in R$, referred to in this paper as points. The algorithm should then automatically label each mesh $P_k \in M \backslash R$, by assigning the label 1 to the points representing the target component and 0 to the remaining points.

### 3.1. Component Detection

In the approach proposed in this paper, object components are detected by segmenting the object's surface into convex and concave segments. This segmentation can be performed using the method proposed in [16]. These segments represent component proposals. The segmentation is performed by segmenting the model mesh into planar patches using the method applied in [17] and aggregating these patches according to the convexity criterion. In this paper, the term concave surfaces is used to denote inverted convex surfaces, i.e., convex surfaces with opposite local surface normals, as illustrated by Figure 1. Each segment of each model in a model database is assigned a unique ID representing a pair of indexes $(i, k)$ where $i$ denotes the segment index and $k$ the model index. One semantic component can be represented by multiple convex or concave segments. For example, the mug handle shown in Figure 2 is represented by one convex and one concave surface.

Since the segmentation of certain shapes can be ambiguous, some additional segments obtained by merging the original segments are created, in order to cover a variety of possible segmentation variants. The algorithm applied for this segment merging is described in Appendix A.

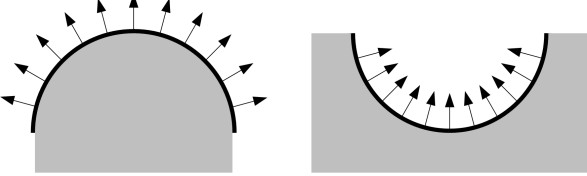

**Figure 1.** An example of a convex (**left**) and a concave (**right**) surface with surface normals denoted by arrows.

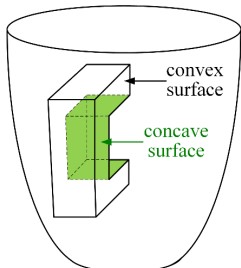

**Figure 2.** Representation of a mug handle by one convex and one concave surface.

### 3.2. Component Association Graph

The method for establishing associations between segments of the objects of the same class, proposed in this paper, is based on the component association graph (CAG). Nodes of the CAG represent segments of all models from the database, connected by edges with the assigned semantic correspondence likelihood measure (SCLM). Each segment of the query object is, thus, connected with $n$ nodes with the greatest SCLM values, referred to in this paper as the nearest neighbors. The number of nodes is limited in order to reduce the computational complexity, which is especially important in the case of large databases. The SCLM consists of the segment shape, size, position, and neighborhood similarity measure. Three methods for establishing the final associations between segments and the target component based on the constructed CAG are proposed in Section 5. As a result of this association process, each segment in the model database is assigned a label with a value of one, if this segment is associated with the target component, or a value of zero otherwise.

## 4. Semantic Correspondence Likelihood

The likelihood that a query object segment $C_j^Q$ and the $k^{\text{th}}$ model segment $C_{ik}^M$ represent semantically corresponding components is assessed by the SCLM computed by:

$$y_{ijk} = y_{ijk}^C + w_N y_{ijk}^N, \tag{1}$$

where $y_{ijk}^C$ represents segment shape, size, and position similarity, $y_{ijk}^N$ represents the segment neighborhood similarity, and $w_N$ is a weighting factor. The computation of the shape, size, position, and neighborhood similarity is described in the following subsections.

### 4.1. CTI Descriptor

The computation of the SCLM is based on the CTI descriptor proposed in [3]. In this subsection, a brief description of this descriptor is provided. Let $A$ be a set of different unit vectors $a_m \in \mathbb{R}^3, m = 1, ..., n_d$ representing standardized normals. Furthermore, let us consider a set of all convex polyhedrons such that each face of the polyhedron is perpendicular to one of the vectors $a_m \in A$. The set $A$ is referred to as a convex template, and each convex polyhedron belonging to the considered polyhedron set is referred to as a convex template instance (CTI). CTI is uniquely defined by a convex template $A$ and a vector $d = [d_1, d_2, ..., d_{n_d}]$, where $d_m$ represents the distance between the $m^{\text{th}}$ polyhedron face and the origin of the object RF. This vector represents the CTI descriptor. The approach proposed in this paper requires that for each unit vector $a_m \in A$, there exist its opposite vector $a_{\overline{m}} \in A$. The CTI descriptor was originally designed for fruit recognition [3]. Later, it was applied for the alignment of similar shapes with the purpose of object classification on depth images [18]. In [16], the CTI descriptor was applied for solving the shape instance detection problem. A CTI descriptor is computed for each object segment. Four examples of objects represented by CTIs are shown in Figure 3.

### 4.2. Segment Shape, Size, and Position Similarity

The shape, size, and position similarity of two segments is measured by comparing their CTI descriptors. In order to reduce computational complexity, the CTI descriptors, $d$, are projected onto a lower dimensional latent space, as proposed in [19]. Thereby, descriptors $q$ of $n_q < n_d$ elements are obtained by:

$$q = O^T d, \tag{2}$$

where $O$ represents an orthonormal basis defining a latent space computed by performing the principal component analysis (PCA) of the CTI descriptors $d$ extracted from the segments of a training set. Another reason for computing the latent vectors is to decouple the shape from the position information. The first three elements of $q$, denoted in this paper by $q^t$, represent the position of the segment in the object RF, while the other 21 elements, denoted in this paper by $q^s$, describe its shape and size. Let us

consider a model database $M$ of $n_M$ models. The segments of the $k^{\text{th}}$ model are represented by latent vectors $q_{ik}^M, i = 1, ..., n_{M,k}$. Analogously, segments of the query object are represented by the latent vector $q_j^Q, j = 1, ..., n_Q$.

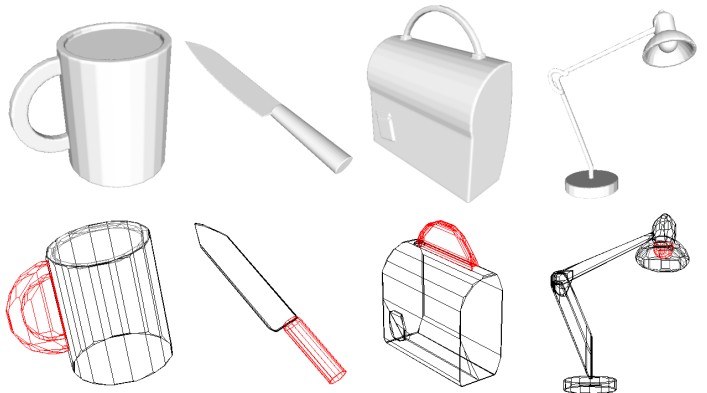

**Figure 3.** Representation of objects by convex template instances (CTIs).

The similarity between two segments taking into account their position ($t$), shape ($s$), and size ($a$) can be measured by Gaussian function:

$$y_{ijk}^C = \exp\left(-\frac{1}{2}\left(\frac{e_{ijk}^t}{\sigma_t^2} + \frac{e_{ijk}^s}{\sigma_s^2} + \frac{e_{ijk}^a}{\sigma_a^2}\right)\right), \tag{3}$$

where $\sigma_t$, $\sigma_s$, and $\sigma_a$ represent parameters that define the contribution of the difference between the segment position, shape and size to the total similarity measure, respectively. The values of the algorithm parameters used in the experiments reported in Section 7 are given in that section. Translation, shape, and scale differences are computed by the following three equations.

$$e_{ijk}^t = \frac{||q_j^{Q,t} - q_{ik}^{M,t}||^2}{||q_j^{Q,s}||||q_{ik}^{M,s}||}, \tag{4}$$

$$e_{ijk}^s = 1 - \left(\left(\frac{q_j^{Q,s}}{||q_j^{Q,s}||}\right)^T \left(\frac{q_{ik}^{M,s}}{||q_{ik}^{M,s}||}\right)\right)^2, \tag{5}$$

$$e_{ijk}^a = \frac{\left(||q_j^{Q,s}|| - ||q_{ik}^{M,s}||\right)^2}{||q_j^{Q,s}||||q_{ik}^{M,s}||}. \tag{6}$$

The norm of vector $q^s$ represents a measure of the segment size. In (4), the position difference is normalized by the size of the segments, therefore allowing bigger segments to have greater distance in order to achieve the same similarity measure values. Equation (5) contains a scalar product of two unit vectors. The greater the value of the scalar product, the more similar the shapes are. Equation (6) represents a segment size difference measure.

### 4.3. Neighborhood Similarity

Let us assume two segments $C_i$ and $C_j$, described by CTI descriptors $d_i$ and $d_j$, belonging to the same model. In order to describe the geometrical arrangement of segments in the model, we introduce

the TRED that describes topological relations between two segments. The TRED represents a tuple $T(C_i, C_j) = (\mu_{ij}, v_{ij}, \sigma_{ij}^v)$. The relation type coefficient, $\mu_{ij}$, is computed by:

$$\mu_{ij} = \min_{m=1,\ldots,n_d} \rho_{ijm}, \tag{7}$$

where:

$$\rho_{ijm} = \frac{d_{im} + d_{j\overline{m}}}{d_{im} + d_{i\overline{m}}}. \tag{8}$$

In Equation (8), $m$ and $\overline{m}$ represent the indexes of two opposite unit vectors $a_m, a_{\overline{m}} \in A$. Five types of topological relation between two segments $C_i$ and $C_j$, defined by $\mu_{ij}$, are considered:

Type 1: $C_j$ contains $C_i$:

$$\mu_{ij} \geq 1 \wedge \mu_{ij} > \mu_{ji}$$

Type 2: $C_i$ and $C_j$ are identical:

$$\mu_{ij} = \mu_{ji} = 1$$

Type 3: $C_i$ and $C_j$ intersect:

$$0 < \mu_{ij} < 1 \wedge 0 < \mu_{ji} < 1$$

Type 4: $C_i$ touches $C_j$:

$$\mu_{ij} = 0$$

Type 5: $C_i$ and $C_j$ are disjoint:

$$\mu_{ij} < 0.$$

The relation type between two segments is denoted in this paper by $type(T(C_i, C_j))$. Three of these relations are illustrated by Figure 4: a segment $C_i$ is inside a segment $C_j$ (left); two segments $C_i$ and $C_j$ intersect (middle); and a segment $C_i$ touches a segment $C_j$ (right).

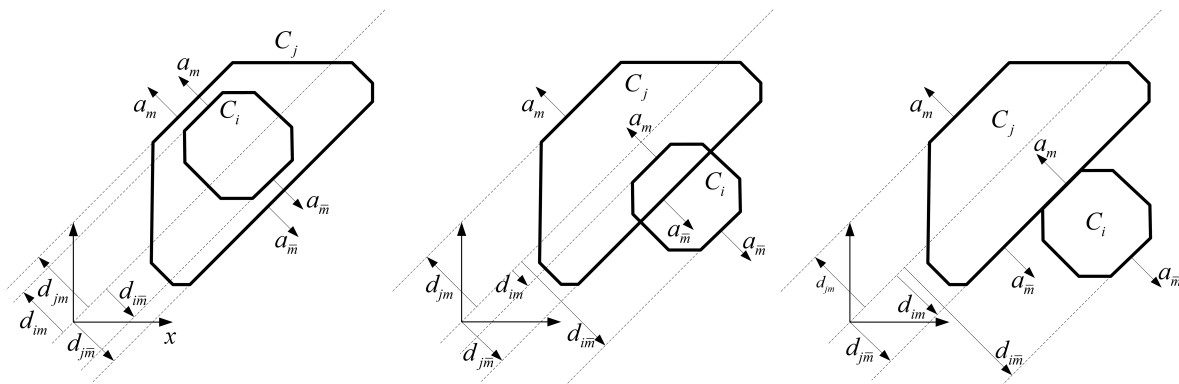

**Figure 4.** Topological relation descriptor (TRED). The CTI descriptor elements $d_{xm}$ that are denoted with arrows directed opposite the corresponding vector $a_m$ have negative values.

If two segments are touching (Type 4), $v_{ij}$ and $\sigma_{ij}^v$ are computed by:

$$v_{ij} = \frac{\sum\limits_{m=1}^{n_d} w_{ijm} a_m}{\left\| \sum\limits_{m=1}^{n_d} w_{ijm} a_m \right\|}, \tag{9}$$

and:

$$\sigma_{ij}^v = \sqrt{\frac{\sum_{m=1}^{n_d} w_{ijk} \arccos^2 v_{ij}^T a_m}{\sum_{m=1}^{n_d} w_{ijk}} + \sigma_{v0}^2}, \tag{10}$$

where:

$$w_{ijm} = \exp\left(-\frac{1}{2}\left(\frac{\mu_{ij} - \rho_{ijm}}{\sigma_\rho}\right)^2\right). \tag{11}$$

Vector $v_{ij}$ is approximately orthogonal to the plane that separates $C_i$ and $C_j$, and it is directed from $C_i$ to $C_j$. Its purpose is to describe the touching direction. This vector is computed as a weighted average of all vectors $a_m$, where vectors corresponding to smaller values $\rho_{ijm}$ have greater weights. Value $\sigma_{ij}^v$ describes the uncertainty of vector $v_{ij}$, where $\sigma_{v0}$ is a constant that models the uncertainty due to the measurement noise. TREDs are computed for every pair of model segments. A segment $C_j$ is a neighbor of a segment $C_i$ if $type(T(C_i, C_j))$ is 1 or 4. In order to make the method robust to noise, the definitions of the topological relation types are relaxed by introducing a suitable tolerance. As a consequence of this relaxation, the topological relation of Type 2 includes not only identical segments, but also segments that occupy approximately the same space.

Furthermore, if two segments are of different convexity types (convex or concave), then they are considered to be neighbors only if their topological relation is of Type 1 or Type 2.

The neighbors of each segment are grouped into clusters according to the associated TRED descriptors. These clusters are denoted in this paper by $\Gamma$, and the cluster set assigned to a segment $C_p^X$ is denoted by $N_p^X$, where $X$ stands for the query ($Q$) or representative ($M$) model. Two neighbors of $C_i$ are grouped into a cluster only if their topological relation to $C_i$ is of the same type. In the case of the Type 4 relation, two neighbors $C_j$ and $C_k$ can be grouped in the same cluster if their vectors $v_{ij}$ and $v_{ik}$ are similar. Vectors $v_{ij}$ and $v_{ik}$ are considered to be similar if the angle between them is $\leq 60°$. In the case of the Type 1 relation, all neighbors $C_j$ such that $\mu_{ij} > \mu_{ji}$ are grouped in one cluster and all neighbors $C_j$ such that $\mu_{ij} < \mu_{ji}$ are grouped in another cluster. All neighbors of different convexity types than $C_i$ whose topological relation is of Type 2 are grouped into one cluster.

Similarity between the neighborhood of a segment $C_j^Q$ of a query object and a segment $C_{ik}^M$ of the $k^{\text{th}}$ model is measured by matching their neighbor clusters. Let $C_m^Q$ and $C_{lk}^M$ be neighbors of $C_j^Q$ and $C_{ik}^M$, respectively. The similarity between these two segments is measured by:

$$y_{lmk}^n = \varepsilon_{lmk} \exp\left(-\frac{1}{2}\left(\frac{e_{lmk}^s}{\sigma_s^2} + \frac{e_{lmk}^a}{\sigma_a^2}\right)\right), \tag{12}$$

where $\varepsilon_{lmk}$ is a binary variable, which has a value of one only if:

$$type(T(C_j^Q, C_m^Q)) = type(T(C_{ik}^M, C_{lk}^M)),$$

or a value of zero otherwise. Furthermore, two neighbors $C_m^Q$ and $C_{lk}^M$ of Type 4 are matched only if vectors $v_{jm}$ and $v_{ilk}$ of the associated TREDs have sufficiently similar directions taking into consideration their uncertainties described by $\sigma_{jm}^v$ and $\sigma_{ilk}^v$. Otherwise, $\varepsilon_{lmk} = 0$.

The similarity of two neighbor clusters $\Gamma_{rj}^Q \in N_j^Q$ and $\Gamma_{pik}^Q \in N_{ik}^M$, where $p$ and $r$ represent cluster indexes, is measured by the similarity of the shape and size of the two most similar segments belonging to these two clusters. The similarity measure of two clusters is computed by:

$$y^\Gamma(\Gamma_{rj}^Q, \Gamma_{pik}^M) = \max_{\substack{C_m^Q \in \Gamma_{rj}^Q \\ C_{lk}^M \in \Gamma_{pik}^M}} y_{lmk}^n.$$

The neighborhood similarity measure is computed by applying the following procedure. First, an initially empty set $B_{ijk}^N$ is created. Then, the pair of clusters with the highest value $y^\Gamma$ is identified

and stored in $B_{ijk}^N$. This step is repeated for all remaining clusters until no more cluster pairs can be formed. Finally, the resulting set $B_{ijk}^N$ is used to compute the neighborhood similarity measure according to the following formula:

$$y_{ijk}^N = \sum_{(\Gamma,\Gamma')\in B_{ijk}^N} y^\Gamma(\Gamma,\Gamma').$$

## 5. Associating Segments with the Target Component

There are three proposed variants of assigning the target component to a query object segment based on the constructed CAG. Note that the construction of CAG is independent of the applied variant.

### *Direct Segment Association*

The query object segment is assigned the ID of the nearest neighbor in the CAG representing a segment of a representative object.

### *Object-Constrained Association*

The object-constrained association approach compares the query model with each representative model. Each segment of the query model is associated with every component of the representative model by a greedy search. The association with the greatest SCLM is detected, and the query segment is assigned the ID of the associated representative model component. That query segment is omitted from the further search, and the search continues until all query segments are associated with corresponding representative object segments. An object matching score between the query and the representative model is computed as the sum of SCLMs between the associated query model segments and representative model segments that represent the target component. Query model segments are finally assigned the IDs of the associated segments of the representative model with the greatest object matching score.

### *MST-Based Association*

A root node is added to the CAG, which does not represent any segment. Then, a minimum spanning tree (MST) is constructed with the constraint that all segments of the representative objects are directly connected to the root node. Hence, each representative object segment of all representative models spreads one subtree. Segments of the query object are assigned the ID of the representative model segment in the same subtree.

At the end of any of the three proposed procedures, each query object segment is assigned the ID of a representative object segment. Finally, the query object segments inherit the labels of the associated representative object segments. The first and the third variant associate each query object segment with the ID of a corresponding representative object segment, independent of the labels of the representative object segments. Hence, these two methods can be performed before the target component is annotated on the representative objects.

## 6. Selection of Representative Objects

Let us consider a model database $M$ representing a set of models $M_k$ of objects belonging to the same object class. Each model $M_k$ is represented by a 3D mesh. As previously explained, every mesh is segmented into convex and concave segments, and each of these segments is represented by a CTI descriptor. In order to facilitate the explanations of the proposed approach, a model $M_k$ will be represented by a sequence of point sets $M_k = (C_{1k}, \ldots, C_{n_M,k}, P_k)$, where $C_{ik}$ is the $i^{\text{th}}$ model segment and $P_k$ is a set of 3D points lying on the surface of the $k^{\text{th}}$ model. For the computational efficiency, these points can be obtained by sub-sampling the mesh vertices. The similarity of the $k^{\text{th}}$ and the $l^{\text{th}}$

object is assessed by measuring the distances between the points of the $k^{\text{th}}$ model to the CTIs of the $l^{\text{th}}$ model and opposite. The object similarity measure is computed by:

$$y_{kl}^O = y_{lk}^O = z_{kl}^O \cdot z_{lk}^O, \tag{13}$$

where:

$$z_{kl}^O = \frac{1}{|P_k|} \sum_{p \in P_k} \exp\left(-\frac{1}{2} \frac{\delta_{pM}^2(p, M_l)}{\sigma_p^2}\right) \tag{14}$$

$$\delta_{pM}(p, M_l) = \min_{C_{jl} \in M_l} (\delta_{pC}(p, C_{jl})) \tag{15}$$

$$\delta_{pC}(p, C_{jl}) = |\max_{m=1,\ldots,n_d} (a_m^T p - d_{jlm})| \tag{16}$$

Parameter $\sigma_p$ in (14) is an experimentally determined constant. If $C_{jl}$ is a concave segment, then the *min* operation is used in (16) instead of *max*. Note that $y_{kl}^O \in [0, 1]$.

Set $R$ of the representative objects is selected by a greedy procedure that maximizes the similarity between $R$ and all objects $M_k \in M$. The similarity between an object model $M_k \in M$ and set $R$ is defined as the similarity between this model and the most similar model $M_r \in R$. This similarity can be measured by a value $y_k^R$ computed by:

$$y_k^R = \max_{M_r \in R} y_{kr}^O.$$

The similarity between $M$ and $R$ can be measured by value $y^{MR}$ representing the total sum of values $y_k^R$, i.e.,

$$y^{MR} = \sum_{k=1}^{|M|} y_k^R.$$

In the approach proposed in this paper, the set of representative objects $R$ is selected by an iterative procedure, where in each iteration, a model that maximizes $y_{MR}$ is selected until a predefined number of representative objects $n_R$ is selected.

## 7. Experimental Evaluation

The proposed approach was experimentally evaluated using the ShapeNet 3D model database [5]. A part of this dataset is dedicated to testing methods that segment 3D models into semantic parts. This dataset consists of 16 subsets, each representing one object class. Each of these subsets is subdivided into training, validation, and test subsets, where the training subsets are significantly larger than the validation and test subsets. The dataset was originally designed to be used in the following way: the evaluated method should be trained using a manually annotated training dataset and the associated validation dataset and tested using the corresponding test dataset. However, in this paper, we investigated a different paradigm. We wanted to test the ability of the proposed approach to (i) segment unannotated 3D models of a particular object class into segments and establish associations between these segments according to the similarity of their shape, size, and geometric arrangement and (ii) to use the established associations to identify a user selected component in all models given a small set of manually annotated representatives. In order to automatize the experiments that require a user selection, instead of manually annotating the representative set in each experiment, we used the ground truth annotations available for every model in the dataset.

Since the target application of the proposed approach was facilitating robotic operations, we selected six object classes from the considered dataset, which could be associated with an exactly defined robot task: mugs, knives, bags, lamps, caps, and laptops. For the mugs, knives, and bags, the target component was the handle for the robot grasping task. For the lamps, the target component

was the light bulb for the replacement task. The target component of the cap was the peak for the dressing task, and the target component of the laptop was the screen for the opening and closing task.

In the case of all classes except the lamps, we used the original annotations of the ShapeNet dataset, where the object handle, peak, and screen were annotated as one of the semantic categories. In the case of lamps, we selected a subset of 210 lamps with clearly distinguishable light bulbs and annotated the light bulbs manually.

The following experimental procedure was applied to each of the six considered object classes, with the parameter values given in Table 1.

1.  Every model was segmented into convex and concave segments using the approach proposed in [16].
2.  Segment merging was performed by the procedure described in Appendix A.
3.  Every segment was represented by a CTI descriptor. Analogously to [3,19], we used a convex template consisting of $n_d = 66$ unit vectors uniformly distributed on the unit sphere.
4.  The TRED descriptor, described in Section 4, was computed for every segment pair of every model.
5.  The neighbor clusters were identified and assigned to every segment.
6.  The CTI descriptor of every segment was projected onto the corresponding latent space by (2).
7.  The SCLM proposed in Section 4 was computed between each of two segments of different models in the model set.
8.  A CAG was created. The nodes of this graph represented all segments of all models in the model set. Each node was connected to at most 100 most similar segments, according to the SCLM computed in Step 7.
9.  A small number of representative objects was selected by the method proposed in Section 6. This number was ≤10% of all models in the considered model set.
10. A target component was annotated on the representative objects. In a practical application, this step would be performed manually by a human expert. In the reported experiments, the ground truth annotations, which were available for all models, were assigned to the representative object models.
11. All segments of all remaining models were automatically annotated using the CAG created in Step 8 and the three methods proposed in Section 5: direct segment association (DSA), object-constrained association (OCA), and MST-based association (MST).
12. The result obtained for every model was compared to the ground truth using the intersection over union (IoU) performance index.

Steps 1–8, the computation of object similarity in Step 9, and the creation of MST in Step 11 were implemented in C++. Step 12 and the rest of the Steps 9 and 11 were implemented in MATLAB.

**Table 1.** Parameter values.

| $n_d$ | $n_q$ | $\sigma_t$ | $\sigma_s$ | $\sigma_a$ | $\sigma_{v0}$ | $\sigma_p$ | $\tau_{tr}$ | $\tau_v$ |
|---|---|---|---|---|---|---|---|---|
| 66 | 24 | 2.4 | 0.132 | 0.707 | 0.1 | 0.025 | 0.333 | 2 |

The computation of the latent vector, performed in Step 6, required an orthonormal basis defining a latent space. This orthonormal basis was represented by an $n_d \times n_q$ matrix $O$. This matrix was computed by the training procedure proposed in [19], where simpler CTI descriptors were used instead of the more complex VolumeNet descriptors, as proposed in [20]. The training was performed using the 3DNet dataset [21]. A total of 351 3D meshes of objects belonging to 10 classes were segmented into convex and concave surfaces, and a CTI descriptor was computed for each surface. Each CTI descriptor represented a point in an $n_d$-dimensional space. The obtained descriptors were collected in two subsets, one representing convex segments and the other concave segments. Matrix $O$

was computed for each descriptor set using the method based on the principal component analysis (PCA) proposed in [19]. For both convex and concave segments, we generated a latent space of $n_q = 24$ dimensions. This number of dimensions was chosen as the minimum number of the first principal components, such that the variances of the remaining principal components were $\leq 10^{-4}$ for both the convex and concave segment set.

The ground truth component annotation was available in the ShapeNet dataset for every model as a set of sample points with assigned labels. Since the proposed approach computed segment associations, the segments had to be associated with the labeled points. Each point was associated with the closest segment. Furthermore, the distance between every point and the CTI of every segment were computed, and every point was associated with a segment if the distance to the CTI of this segment was $\leq 0.001$. This distance was computed by (16). Note that this was not the Euclidean distance, but it was a good approximation, which could be computed very efficiently. The same distance threshold could be used for all objects, since all models in the ShapeNet dataset were scaled to approximately the same size. This point-segment association allowed each point to be associated with more than one segment.

In Step 10 of the experimental procedure, a segment of a representative object was labeled as being part of the target component if more than half of the points it was associated with were annotated. Step 11 annotated the segments of all models from a considered model set, except the representative objects, which were already annotated in Step 10. In order to compute IoU in Step 12, annotations must be transferred from segments to sample points. A point was labeled as belonging to the target component if it was associated with at least one annotated segment.

## 7.1. Results

A few sample results are shown in Figure 5. True positives, i.e., points that were correctly associated with the target component, are depicted by the green color, while false positives, i.e., points that were falsely associated with the target component, are depicted in red color. The false negatives, representing the target component points, which were not associated with the target component by the proposed algorithm, are depicted in blue color.

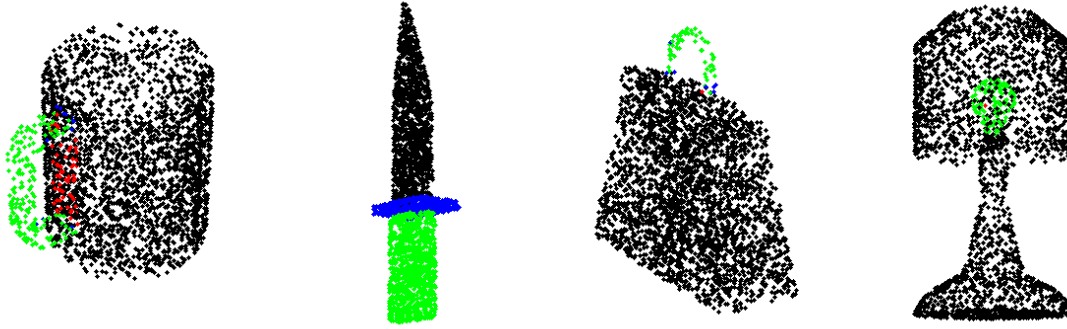

**Figure 5.** Sample results.

The quantitative results obtained by the described experimental procedure are presented in Table 2. Each of the three component association methods were tested with and without using the neighborhood similarity, i.e., with weight $w_N$ equal to one and zero, respectively. The number of representative objects was the greatest integer $\leq 10\%$ of all models in the model database of each object class. The presented IoU values represented the average over all objects of a particular class excluding the representative objects.

**Table 2.** Accuracy of the automatic component annotation. DSA, direct segment association; OCA, object-constrained association; MST, minimum spanning tree.

| | Class | | Mug | Knife | Bag | Lamp | Cap | Laptop |
|---|---|---|---|---|---|---|---|---|
| | total No. of objects | | 184 | 391 | 68 | 210 | 53 | 451 |
| | No. of representatives | | 18 | 39 | 6 | 21 | 5 | 45 |
| IoU | DSA | $w_N = 0$ | 78.50 | 75.40 | 44.10 | 43.10 | 82.50 | 75.30 |
| | | $w_N = 1$ | 77.40 | 73.30 | 50.30 | 57.20 | 84.80 | 77.20 |
| | OCA | $w_N = 0$ | 73.90 | 58.00 | 48.00 | 53.60 | 82.40 | 58.90 |
| | | $w_N = 1$ | 75.10 | 62.40 | 43.60 | 65.40 | 82.40 | 59.30 |
| | MST | $w_N = 0$ | 77.90 | 75.90 | 42.60 | 36.80 | 82.70 | 76.70 |
| | | $w_N = 1$ | 76.30 | 73.20 | 60.90 | 53.00 | 75.90 | 76.10 |

In order to compare the proposed method with the state-of-the-art approaches, the following experiment was executed. For each of the five classes: mugs, knives, bags, caps, and laptops, Steps 1–10 were performed on the training subset, which included the selection and annotation of representatives. Note that only a small representative subset of the training dataset was annotated. The number of annotated representatives was the greatest integer ≤10% of all models in the training subset of each object class. Then, the database was extended (Steps 1–8) by the test dataset; the labels were assigned to the models from this subset; and the accuracy analysis was performed (Steps 11–12). The results of the accuracy analysis performed on the test subset measured by per-class IoU are given in Table 3. At the bottom of the table, the accuracies achieved by the three methods (DSA, OCA, and MST) proposed in this paper are given. Since this paper investigated component association with a small set of representatives, it was not expected to outperform the existing methods extensively trained using large annotated datasets.

**Table 3.** Accuracy (%) comparison with the state-of-the-art methods.

| Method/Class | | Mug | Knife | Bag | Cap | Laptop |
|---|---|---|---|---|---|---|
| SyncSpecCNN [22] | | 92.73 | 86.10 | 81.72 | 81.94 | 95.61 |
| Pd-Network [23] | | 94.00 | 87.25 | 82.42 | 87.04 | 95.44 |
| SSCN [24] | | 95.23 | 89.10 | 82.99 | 83.97 | 95.78 |
| SpiderCNN [25] | | 93.50 | 87.30 | 81.00 | 87.20 | 95.80 |
| SO-Net [26] | | 94.20 | 83.90 | 77.80 | 88.00 | 94.80 |
| PCNN [27] | | 94.80 | 86.00 | 80.10 | 85.50 | 95.70 |
| KCNet [28] | | 94.40 | 87.20 | 81.50 | 86.40 | 95.50 |
| Kd-Net [23] | | 86.70 | 87.20 | 74.60 | 74.30 | 94.90 |
| 3DmFV-Net [29] | | 94.00 | 85.70 | 84.30 | 86.00 | 95.20 |
| RSNet [30] | | 92.60 | 87.00 | 86.40 | 84.10 | 95.40 |
| DGCNN [31] | | 93.30 | 87.30 | 83.70 | 84.40 | 96.00 |
| PointNet [32] | | 93.00 | 85.90 | 78.70 | 82.50 | 95.30 |
| PointNet++ [33] | | 94.10 | 85.90 | 79.00 | 87.70 | 95.30 |
| SGPN [34] | | 93.80 | 83.00 | 78.60 | 78.80 | 95.80 |
| PointCNN [6] | | 95.28 | 88.44 | 86.47 | 86.04 | 96.11 |
| DSA | $w_n = 0$ | 82.14 | 75.53 | 39.67 | 69.59 | 75.54 |
| | $w_n = 1$ | 76.65 | 73.42 | 41.66 | 74.76 | 74.53 |
| OCA | $w_n = 0$ | 75.74 | 56.35 | 36.24 | 81.45 | 59.67 |
| | $w_n = 1$ | 76.47 | 59.18 | 40.34 | 80.28 | 59.95 |
| MST | $w_n = 0$ | 77.68 | 75.59 | 34.92 | 75.57 | 76.74 |
| | $w_n = 1$ | 75.80 | 64.78 | 44.01 | 72.32 | 65.91 |

Let us assume a database with the total number of objects as stated in Table 4. The running times of the automatic component annotation with the included neighborhood similarity ($w_N = 1$) in

the SCLM are reported in Table 4. The average execution time of every step per object is presented. Furthermore, the total execution time of all steps required for expanding the model database with a new query object is provided. It was assumed that before the expansion, the database contained the total number of objects given in Table 4 minus one. The average running time per model is reported for Steps 1–8, implemented in C++, and for Steps 11 and 12, implemented in MATLAB. In the case of the MST method, the execution time of Step 11 included computation of MST implemented in C++. The representative selection time, Step 9, was required only when the representative models were selected. In the case where the database was extended without annotating a new representative object, this step was not performed. Thus, the execution times of this step are presented separately from the other steps. Furthermore, Step 10, which was performed by a human expert, is not considered in Table 4. The experiments were executed on a PC with an Intel Core i7-4790 3.60GHz processor and 16 GB of installed RAM memory running Windows 10 64-bit OS.

**Table 4.** Running time of the automatic component annotation.

| Class | | | Mug | Knife | Bag | Lamp | Cap | Laptop |
|---|---|---|---|---|---|---|---|---|
| total no. of objects | | | 184 | 391 | 68 | 210 | 53 | 451 |
| no. of representatives | | | 18 | 39 | 6 | 21 | 5 | 45 |
| average step execution time per object [ms] | C++ | steps 1–6 | 825 | 1112 | 1484 | 917 | 702 | 288 |
| | | steps 7–8 | 19 | 70.6 | 36.2 | 62.4 | 3.8 | 14.6 |
| | MATLAB steps 11–12 | DSA | 4.7 | 10.6 | 5.6 | 8.2 | 2.1 | 3.5 |
| | | OCA | 109.7 | 667.5 | 11.9 | 199.1 | 4.6 | 383.4 |
| | | MST | 52.2 | 106.7 | 14.9 | 45.4 | 5.3 | 72.5 |
| total time per object [ms] | | min | 849 | 1193 | 1526 | 988 | 708 | 306 |
| | | max | 954 | 1850 | 1535 | 1179 | 711 | 686 |
| representative selection time [ms] | C++ | step 9 | 59,847 | 198,611 | 12,249 | 82,930 | 4873 | 314,175 |
| | MATLAB | step 9 | 10 | 26 | 2 | 4 | 1 | 35 |

In order to investigate how the component association accuracy depended on the number of representative objects, we performed a series of experiments, where we varied the number of representatives from one to the value shown in Table 2. The results of this analysis are shown in Figure 6. The experiments were performed for all three component association methods with $w_N = 1$.

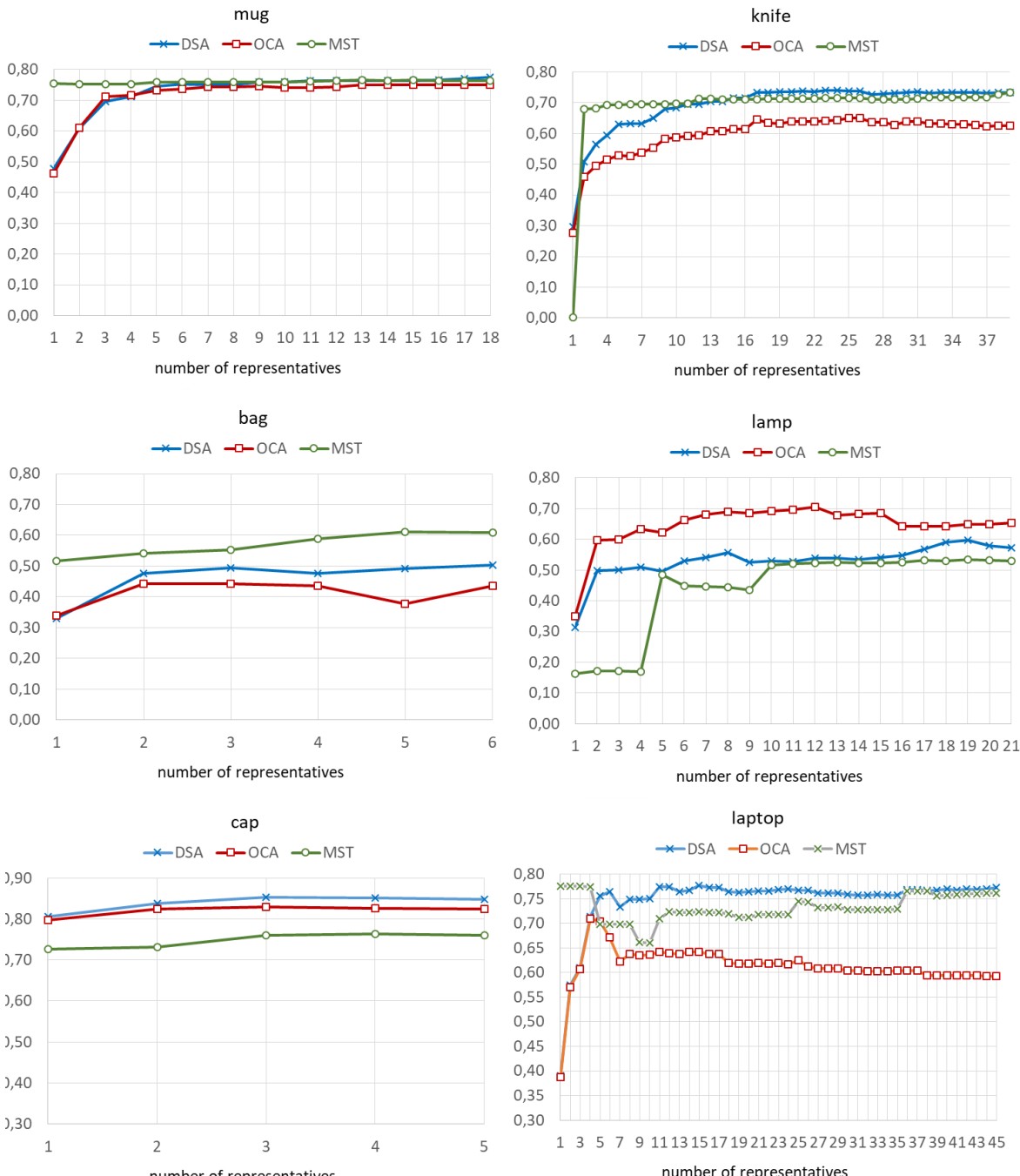

**Figure 6.** Component association accuracy depending on the number of representatives and the applied method measured by IoU.

## 7.2. Discussion

The standard pipeline for semantic object segmentation with standard machine learning or deep neural networks consists of training, validation, and test parts. The training is usually performed on a large subset containing many annotated objects. The time spent for the annotation and training is rarely reported in the research, but it surely requires much time and energy. This experimental analysis investigated how accurate semantic association can be achieved without extensive training and annotation. The reported results proved that only a few annotated representative models were required to achieve the reported accuracy. In the case of simple objects, such as knives, caps, mugs, and laptops, only one or two annotated models were required to achieve an IoU of approximately 0.7.

From the presented experimental analysis, it could be concluded that including neighborhood similarity in the SCLM improved the results significantly in the case of bags and lamps, which had more complex shapes than mugs, knives, caps, and laptops. In the case of mugs, knives, caps, and laptops, which had simple shapes, the information about the topological relations between components did not enhance the ability of the algorithm to identify the target component.

For all classes except the lamps, the MST component association method would be the method of choice, because it provided a relatively high IoU for a small number of representatives, which did not grow significantly with the number of representatives. In the case of mugs, caps, and laptops, a single representative was sufficient to identify the target component in the remaining objects with IoU > 0.70. In the case of knives, the wrong selection of the first representative by the proposed method resulted in a very low IoU. However, with the second representative, an IoU close to 0.7 was reached.

The MST method was expected to provide better results than the DSA in the case where the model database represented a sufficiently dense sampling of the considered object class. We considered a model database to be dense if for any two objects, a sequence of objects could be found where two consecutive objects had sufficiently similar shapes to allow unambiguous segment association. An example of such a sequence is shown in Figure 7. A bag handle of a reference object, shown in the top left image of Figure 7, was correctly associated with the segment represented by green points in the bottom right image of Figure 7. In an ideal case, the processing of such a model database by the MST method would allow correct component association using a single annotated representative object. However, in order for a dataset to be dense and cover large shape variations, it has to be very large. In order to evaluate how the size of the object dataset affected the accuracy of the MST-based component association, the procedure explained in Steps 1–12 was performed for all six classes with different numbers of models in the database, varying from 25% to 100% of the total No. of objects given in Table 2 and with the included neighborhood similarity, $w_N = 1$. The results are given in Table 5.

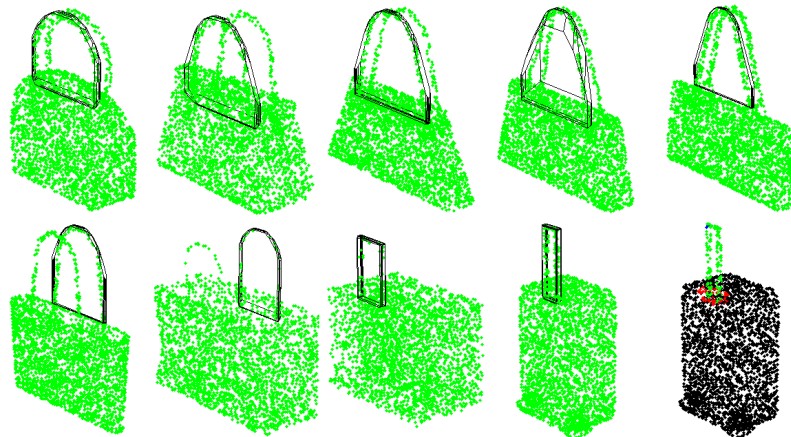

**Figure 7.** An example of the dense database with similar consecutive objects.

**Table 5.** Accuracy of the component annotation by MST with different numbers of objects in the database.

| Percent of Database/Class | Mug | Knife | Bag | Lamp | Cap | Laptop |
|---|---|---|---|---|---|---|
| 25% | 0.721 | 0.755 | 0.400 | 0.080 | 0.706 | 0.762 |
| 50% | 0.747 | 0.726 | 0.419 | 0.177 | 0.732 | 0.694 |
| 75% | 0.752 | 0.749 | 0.604 | 0.400 | 0.793 | 0.739 |
| 100% | 0.763 | 0.732 | 0.609 | 0.530 | 0.759 | 0.761 |

Although the proposed method provided interesting results for a small numbers of representative objects, it did not achieve very high accuracy even when a high percentage of annotated representative objects was used. One of the reasons was that the proposed approach relied on a segmentation method

that provided segments that often crossed the boundary of the target component. This was noticeable in the case of mug and bag handles, where the concave segment representing the inner side of a handle extended to the body of the object, resulting in false positives, as illustrated by the example shown in the leftmost image of Figure 5, where the red points were falsely assigned to the handle.

The goal of the proposed approach was to reduce the annotation effort and allow an easy expansion of the model dataset by new models in real time. The execution times provided in the Table 4 indicated that expanding the model database with a new model, recomputation of the CAG, and establishing the component-segment correspondences required between approximately 0.3 and 1.9 seconds, depending on the complexity of the object structure, the chosen method, and the number of objects already in the model dataset. Thus, the proposed method is usable for real time robotic applications. The active learning framework for annotating massive geometric datasets [8] reported the average time required for preprocessing and annotation per shape of approximately one minute for a database of 400 objects. This time was expected to scale linearly with the size of the database. Furthermore, the analysis of a new shape, in a method for converting geometric shapes into hierarchically segmented labeled parts, proposed by [9], typically took about 25 seconds. Comparing with these two methods, which also propagated labels from a small subset to a large dataset, the method proposed in this paper is significantly faster.

## 8. Conclusions

The aim of the paper was to investigate how accurate semantic association can be achieved without extensive training and annotation of a large amount of data and if such an approach could be time effective for real-time applications. The target application of the proposed approach was in robotics, where it could be used in combination with an object detection and 3D reconstruction module. The task of this module would be to detect objects of a particular class in an RGB-D image or a LiDAR scan and to reconstruct their full 3D model. The proposed approach could then be used to associate the components of the reconstructed object, with the corresponding components of representative object models. In order to be applicable in real-world scenarios, the robot perception system must be able to cope with cluttered scenes, where the target objects are partially visible and appear in various poses. Augmentation of the proposed approach with a method that would detect objects of a given class in complex scenes and perform their 3D reconstruction is our first choice for the continuation of the presented research.

Since the application of neural networks in semantic segmentation and object classification was justified by the reported accuracy in related research, training a neural network for general-purpose component segmentation and matching is an interesting subject for future research. In order to apply a neural network for solving the problem defined in this paper, a neural network should be trained using various datasets in order to learn a generic criterion for possible semantic association between model components. After being trained, the network would be applied for detection and association of the components of another unannotated dataset.

In this paper, a method for semantic association of the object components with the relatively small number of the annotated representative objects was proposed. The method was based on construction of the component association graph, whose nodes represented the object segments. Segments were approximated by convex polyhedrons. The CAG edges were assigned a semantic correspondence likelihood measure between the segments, which took into consideration both segments and their neighborhood similarity. The final association was performed by one of the three proposed methods based on the component association graph. The evaluation of the proposed approach on a subset of the ShapeNet Part benchmark dataset yielded interesting results, which indicated that with a rather small number of annotated representatives, the identification of a target component with accuracy measured by IoU greater than 0.6 could be achieved in only a few seconds.

**Author Contributions:** Conceptualization, R.C.; data curation, P.D.; formal analysis, P.D. and R.C.; funding acquisition, R.C.; investigation, P.D. and R.C.; methodology, P.D. and R.C.; project administration, R.C.; software, P.D., I.V., and R.C.; supervision, R.C.; validation, R.C.; visualization, R.C.; writing, original draft, P.D. and R.C.; writing, review and editing, P.D. and R.C. All authors have read and agreed to the published version of the manuscript.

**Funding:** This work has been fully supported by the Croatian Science Foundation under the project number IP-2014-09-3155.

**Acknowledgments:** We are especially grateful to Li Yi from Stanford University for kindly providing us the mapping between ShapeNet dataset meshes and point cloud model names and helping us with other dilemmas regarding ShapeNet.

**Conflicts of Interest:** The authors declare no conflict of interest. The funders had no role in the design of the study; in the collection, analyses, or interpretation of data; in the writing of the manuscript; nor in the decision to publish the results.

## Appendix A. Segment Merging

The criterion used to select the candidates for merging is based on proximity of two segments and convexity of their union. Two segments $C_i$ and $C_j$ satisfy the proximity criterion if:

$$\min(\mu_{ij}, \mu_{ji}) \geq -\tau_{prox,1},$$

where $\tau_{prox,1} = 0.2$ is an experimentally determined threshold. The convexity of the union of two segments is evaluated by computing its convex hull and counting the outlier points, i.e., points that do not lie on the convex hull within a predefined threshold. The outlier percentage is defined by:

$$\eta(C_i, C_j) = \frac{|C_i \cup C_j| - |inliers(C_i \cup C_j, CH(C_i \cup C_j))|}{\min(|C_i|, |C_j|)},$$

where $|X|$ denotes the cardinality of a set $X$, $CH(X)$ denotes the convex hull of a set $X$, and $inliers(X, Y)$ denotes the inlier set. The inlier set contains all points $p \in X$ that are close to the surface of a convex polyhedron $Y$. Furthermore, $p \in inliers(X, Y)$ only if the local surface in the vicinity of $p$ is similarly oriented as the surface of $Y$ in the vicinity of $p$. This is evaluated by the following criterion based on the point-to-plane distance between $p$ and the tangential plane of $Y$ with the normal parallel to the local surface normal of $p$:

$$\max_{p' \in V_Y} (n_p^T(p' - p)) \leq \tau_{prox,2},$$

where $n_p$ is the local surface normal of $p$, $V_Y$ is the set of vertices of $Y$, and $\tau_{prox,2}$ is an experimentally determined threshold. In the experiments reported in Section 7, $\tau_{prox,2}$ was set to 5% of the radius of the model's bounding sphere, where an approximate minimum bounding sphere was computed by the algorithm proposed in [35]. If $\eta(C_i, C_j) < \tau_{new}$, then a new segment is created from the union of $C_i$ and $C_j$. If $\eta(C_i, C_j) < \tau_{merge} < \tau_{new}$, then the new segment replaces $C_i$ and $C_j$, i.e., the two original segments are rejected. The values of thresholds $\tau_{new}$ and $\tau_{merge}$ used in the experiments reported in Section 7 were 0.2 and 0.05, respectively. This merging process was applied not only to the original segments, but also to the new segments created by merging the original segments. In that case, the candidates for merging were always one original and one new segment. If one candidate was an original segment $C_i$ and another was a new segment composed of original segments $C_j$, then the proximity and convexity criterion were evaluated for each pair of segments $(C_i, C_j)$.

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
