# Peer review of "Semantic Component Association within Object Classes Based on Convex Polyhedrons"

_applsci, doi:10.3390/app10082641_

Round 1
Reviewer 1 Report
Essential Comments
The reviewed manuscript is focused on a problem of semantic association of objects of a given class, which has important applications in robotics and in visual data recognition. The Authors propose a new approach based on construction of the component association graph (CAG), whose nodes represent the object segments. Segments are approximated by convex polyhedrons. Three methods of the mentioned associating using CAG are proposed and investigated. Also, an original Topological Relation Descriptor (TRED), which is dedicated to 3D object models is demonstrated.
The paper contains an original and valuable research. Theoretical background and experimental verification are presented. Experiments were performed for 4 different classes of objects from the ShapeNet dataset, which is open accessed, so the experiments can be verified and compared by other scientists. The Authors are aware of limitations of their solution and plan to continue their research using neural network trained using various datasets.
Editorial Comments
The paper is well written and well organized. I have noticed only small editorial mistakes.
- The state of art and description of related works is extensive. Reference list contains 28 positions, however some related methods are described too briefly or not at all. For example, the method used in position [7] on the reference list is not mentioned in the text. The Authors referred to methods presented in positions [21] and [3] but did not put even their names in the text.
- All abbreviations (eg. CTI, MST, CNN) should be explained by their first use in the text.
Reviewer 2 Report
The paper describe a method for a semantic association of objects components based on convex polyhedrons.
The related work part is long and not completely justified, as none of the discussed existing methods has been chosen for comparison in the testing session. In particular, the discussion confuses two extremely different concepts that are segmentation and 6D pose reconstruction.
Going into the overview of the proposed method, there is some confusion in the exposition. Since the beginning, the authors say that the model database is composed of point clouds. However, soon after, all the discussion on the segmentation of objects is based on model surfaces so triangular meshes.
It is not completely clear shy the edges between nodes are limited to the neighbors with greatest SCLM values, for computational reasons?
Beware of using the same terminology for completely different elements, for instance, the M symbol is employed in the text both for representing the database of object models and for the orthonormal basis for latent space reduction.
The manuscript text can be reduced in many passages because in its current form is too long and confusing. For instance, eqs 7 and 8 are not necessary.
In section 6 what dense 3D mesh means? I suggest to explain it or remove the sentence. At line 363 it is written that points are used in the discussion only to facilitate the explanation...still confusing, the implemented system uses point clouds or meshes?
The experimental evaluation seams limited given that it has been tested only with 4 different object categories. How does it extend to other categories? What is the execution time for the entire pipeline? it is useable for robotic applications? Why the remaining 7 classes od the datasets have not been used for evaluation? How the latent space dimensions have been chosen?
The resulting accuracy seams low and no direct comparison with existing methods is presented. At line 477, the authors write the 'SCLM improves the results', with respect to what?
The authors claim that the application of the work is in the robotics field but no consideration has been made about cluttered scenarios, typical of robotic applications.
The content of the manuscript in its current form is borderline.
Reviewer 3 Report
The paper presents an interesting technique of identifying objects based on certain broad indicators with the main aim of reducing the manual effort and time required for training. Some comments below.
- The state of the art or related research section should be rewritten. The current writing is pure reporting style with absolutely no context either wrt the real-world application or linking to the research in this paper. Except for a couple of sentences stating that some method is relevant to the manuscript, no other context is provided. A reader might as well skip this whole section without missing out on any information.
- The writing in related research section should also be improved in terms of explaining the merits and demerits of state of the art. For ex: line 144, the authors state "The development of semantic segmentation methods is encouraged by various challenges such as [21]. Do the authors expect the reader to go through the paper [21] and understand which exact problem the authors have in mind? The same is seen in the whole of this section. In my view, this "related research" section is the poorest part of the paper.
- What is the accuracy of the annotation wrt the size of the training set? If the main objective of the proposed method is to reduce time and manual effort, then this is also critical. It may also be possible that sufficient training data is not available for certain classes. Remarks in this regard will be useful.
- What is the computational effort/time taken for annotation of instances during the testing phase? Considering the application is robotics, fast real-time algorithms are preferred. If training effort is reduced at the cost of increased run time, then this is not industry useful. Please comment on this issue and compare your method to existing methods.
- I would suggest swapping the final two paragraphs of the conclusion section.
Round 2
Reviewer 2 Report
The manuscript has been improved but still needs some minor adjustments.
In particular, 427-431 are not really convincing. Why is not possible to use the same dataset both for training, validation, and testing? If the dataset used for the comparison is not the same the comparison is meaningful.
Please try to perform training with the same dataset with at least 1 or 2 best-performing methods from the one you tested and test again to show the effective difference with the proposed method.
